# Avian Influenza Virus: Comparative Evolution as the Key for Predicting Host Tropism Expansion

**DOI:** 10.3390/pathogens14070608

**Published:** 2025-06-20

**Authors:** Matteo Mellace, Carlotta Ceniti, Marielda Cataldi, Luca Borrelli, Bruno Tilocca

**Affiliations:** 1Department of Health Sciences, University Magna Graecia of Catanzaro, Viale Europa, 88100 Catanzaro, Italy; matteomellace4@gmail.com; 2Department of Public Health, ASL Napoli3 Sud, Corso Alcide De Gasperi, 80053 Castellamare di Stabia, Italy; carlotta.ceniti@aslnapoli3sud.it (C.C.); marielda.cataldi@aslnapoli3sud.it (M.C.); 3Department of Veterinary Medicine and Animal Production, Via Delpino n.1, 80137 Napoli, Italy; luca.borrelli@unina.it; 4Department of Veterinary Medicine, University of Sassari, Via Vienna n.2, 07100 Sassari, Italy

**Keywords:** mutations, spillover, evolution, molecular surveillance

## Abstract

The avian influenza virus poses an emerging public health risk due to its ability to cross the species barrier and infect a broad spectrum of hosts, including humans. The aim of this study was to investigate the molecular mechanisms and evolutionary dynamics underlying the spillover, using a bioinformatics approach to viral sequences. Eight viral proteins involved in the process of adaptation to new hosts were selected, and 156 amino acid mutations potentially associated with interspecies transmission were analyzed. The sequences, obtained from the NCBI Virus database, were aligned with the BLASTP1.4.0 tool and compared through phylogenetic analysis. The results show significant evolutionary proximity between human and animal viral strains, and the identification of shared mutations suggests the presence of conserved mechanisms in spillover. The identification of hosts that share mutations with human strains highlights the potential role of these animals as reservoirs or vectors. This study contributes to the understanding of viral adaptation and provides a starting point for targeted preventive strategies, including molecular surveillance and the development of containment and prevention measures.

## 1. Introduction

Avian influenza (AIV) type A viruses pose a persistent threat to public and veterinary health due to their extraordinary genetic plasticity and ability to infect a broad spectrum of hosts, ranging from wild birds to land and marine mammals, including humans [1]. These viruses, characterized by a segmented genome with single-stranded RNA and negative polarity, are known for their ability to evolve rapidly through antigenic drift and shift mechanisms, favoring the emergence of new, potentially pandemic variants [2,3,4]. Although wild waterfowls are considered the main natural reservoir of AIVs [5], the growing evidence of spillover to domestic and synanthropic species, such as chickens, pigs, cats, and mink, calls for a careful re-evaluation of the ‘mixing vessel’ concept in influenza epidemiology [6]. The integration of clinical, epidemiological, and molecular observations suggests that the interspecific adaptive capacity of AIV is regulated by precise viral determinants, including specific mutations in polymerase proteins [7,8,9], nucleoprotein [10], and surface glycoproteins [11]. Although the literature has fragmentarily described some mutations associated with spillover, systematic models linking such molecular alterations to evolutionary adaptation in mammalian hosts and replicative efficiency in human cells are still lacking. This study aims to fill this gap through advanced bioinformatics and phylogenetic analysis, aimed at identifying amino acid mutations shared between animal and human strains and assessing their potential impact on viral tropism and adaptive evolution. The study makes use of an extensive systematic literature review, an analysis of viral protein sequences in the NCBI Virus database, and the use of computational tools such as BlastP and phylogenetic inference algorithms to characterize mutations distributed across eight major viral proteins. The results show that specific mutations in the functional regions of the PB1, PB2, and PA proteins are frequently shared between zoonotic and human strains, suggesting a convergent evolutionary pattern.

## 2. Materials and Methods

### 2.1. Identification of Proteins and Mutations Potentially Involved in Spillover

A systematic review of the scientific literature of the last ten years was conducted to identify the proteins of the avian influenza type A virus involved in the phenomenon of spillover species, with a focus on mutations that favor adaptation to different hosts or human-to-human transmission. The literature search was performed on the PubMed and Google Scholar databases, using the keywords “Avian Influenza”, “Proteins”, “Mutations” and “Spillover”, and selecting articles that reported experimental evidence or events related to key mutations for interspecies switching. The main viral proteins and specific mutations were identified considering criteria such as belonging to different influenza type A virus strains, the phenotypic effects associated with the mutations, and the role of the proteins in the infection. In particular, the literature search was directed by classifying the documented amino-acid mutations and proteins on different levels of adaptation and expansion of the host spectrum. The mutations considered in this study concern the main viral proteins, both structural and non-structural, which play a fundamental role in the biology and structure of the virus. Each mutation was selected after a careful review of the literature, considering only those variants supported by experimental evidence, both in vitro and in vivo. We prioritized mutations that have been shown to affect viral replication, adaptation to new hosts, and the ability to evade the immune system or facilitate transmission between species, to include only those that are truly relevant in the context of zoonotic transmission.

### 2.2. Protein Sequence Collection and Accession Number Identification

The protein sequences for the identified proteins were downloaded from public databases, using their accession numbers. To download the protein sequences associated with the avian influenza virus type A, the NCBI Virus database was used, drawing on information available in the GenBank database. Viral protein sequences were searched using the search parameter ‘search by virus’ and entering the taxid ‘11320’, the identifier for the *influenza type A virus*, in the search bar. Search filters were selected for the presence of sequences with proven completeness in assembly and nucleotides. The ‘host’ search filter was used for each search performed. The hosts selected for the search were human, chicken, dog, cat, pig, turkey, swan, goose, gull, and duck. For sequences belonging to humans, reference was made to the RefSeq reference database, which provides data and reference sequences for model organisms that are essential to ensure the reliability of the information. The selection of hosts from which the sequences were obtained was conducted based on their epidemiological relevance, as documented in the scientific literature, and their synanthropic propensity—crucial aspects for understanding the dynamics of interaction between the virus and the various hosts. Embracing the One Health framework, the study includes a variety of hosts characterized by different behavioral habits, levels of synanthropy, and habitats of origin. This strategy allows for a more complete analysis of potential spillover pathways and mechanisms of viral adaptation in complex ecological and biological contexts. For each host considered, the sequences for the protein segments of interest associated with each influenza virus strain in the database were selected. Sequence inclusion criteria included geographic distribution, along with sequence completeness and length, with a preference for full-length sequences. The temporal aspect was also considered, with a preference for sequences with more recent publications. In addition, the annotation status of the sequences was also considered, with priority given to reference and well-annotated sequences. This approach made it possible to construct an ad hoc dataset, including viral variants, which will be fundamental for subsequent comparative and phylogenetic analyses, necessary for understanding the evolutionary dynamics of the virus. This multi-host selection was carried out to allow for comparative sequence analysis and to identify mutations associated with interspecific adaptation.

### 2.3. Alignment of Human Protein Sequences and Construction of Phylogenetic Trees

The protein sequences were aligned using the bioinformatics tool BLASTP v 1.4.0 (protein–protein BLAST), exploiting the BLOSUM62 algorithm and keeping the default parameters of the alignment software unchanged. Initially, human reference protein sequences (RefSeq) were aligned against each other to identify any phylogenetic similarities and differences using protein segments belonging to ‘*Influenza A virus* (A/Hong Kong/1073/99(H9N2))’ as queries. The assessment of similarities was carried out considering parameters such as E-value, identity percentage, maximum score, and query coverage. Subsequently, each reference protein sequence was aligned against the sequences of the various host strains, using each of the human reference sequences (RefSeq) as a query: A/Korea/426/1968(H2N2), A/Shanghai/02/2013(H7N9), A/California/07/2009(H1N1), A/Puerto Rico/8/1934(H1N1), A/New York/392/2004(H3N2), and A/Hong Kong/1073/99(H9N2). This allowed for the variations between the different species to be compared. After alignment, phylogenetic trees were constructed using the Fast Minimum Evolution method, with a Max Seq Difference set at 0.85. The Fast Minimum Evolution method was selected for its favorable compromise between processing speed and accuracy in phylogenetic inference, allowing reliable trees to be obtained while minimizing the overall length of the branches. The Max Seq Difference threshold of 0.85 was chosen in line with what was adopted in previous studies on similar viral sequences, to include sequences with an evolutionary divergence of up to 85% and exclude those excessively divergent that could have compromised the quality of the analysis. Although statistical support measures are often used to evaluate the robustness of phylogenetic relationships, in this case, they were not applied considering the predominantly exploratory approach of the analysis and the relative homogeneity of the dataset consisting of sequences belonging to the same viral protein. Nonetheless, these statistical supports could be incorporated into future works for further validation.

### 2.4. Analysis of Phylogenetic Trees and the Presence of Mutations in Aligned Sequences

The phylogenetic trees were carefully examined to analyze, for each host, serotype, and amino acid sequence considered, any evolutionary proximity or distance. Following the evolutionary analysis, pairs of amino acid sequences were selected for the proteins of interest, characterized by significant evolutionary proximity between human references and different hosts. The selection was conducted by identifying, out of the total number of phylogenetic evaluations available for analysis, the amino acid sequence pairs that showed the greatest evolutionary proximity between different viral serotypes. If the sequences belonged to proteins that were highly conserved between the different serotypes (HA and NA), a selection was carried out, in addition to the criteria described above, by assessing which pairs showed 80% or more identity in alignment. The selected sequences were aligned using the bioinformatics tool BLASTP v.1.4.0 (protein–protein BLAST), employing the BLOSUM62 algorithm and maintaining the default parameters of the alignment software. The results of the alignment were subsequently visualized using the graphical viewer NCBI Sequence Viewer v 3.50.0. Using the viewer’s search bar, possible mutations previously identified in the recent scientific literature specific to each of the proteins selected in the preliminary steps were identified within the amino acid sequences. Sequence pairs showing evolutionary proximity between one or more different human and host reference sequences were then selected based on phylogenetic analyses. The previously chosen amino acid mutations were then detected within each of the pairs that fulfilled the selection criteria. After analyzing individual mutations in each pair of sequences, hosts were identified as having one or more mutations present in two or more of the selected proteins shared with one or more human reference genomes.

## 3. Results

### 3.1. Proteins and Mutations Involved in Spillover

The literature search and systematic review of scientific literature over the past 10 years identified 50 scientific reports that provide experimental evidence on the correlation between amino acid mutations and phenomena such as species hopping, adaptation to different hosts, and interhuman transmission. The literature review revealed eight proteins of the avian influenza virus type A associated with these mechanisms: Polymerase Basic Protein 1 (PB1), Polymerase Basic Protein 2 (PB2), Polymerase Acidic (PA), Nucleoprotein (NP), Hemagglutinin (HA), Neuraminidase (NA), Matrix Proteins 2 (M2), and Non-structural Protein 1 (NS1). A total of 156 mutations were found to be involved, distributed among the eight proteins: 26 in PB1, 38 in PB2, 41 in PA, 5 in NP, 32 in HA, 12 in NA, 4 in M2, and 3 in NS1 (Table 1). The amino acid mutations identified in the recent literature have been classified according to their functional relevance. The mutations have been classified into two groups: those located within well-defined functional domains of viral proteins and those associated with a functional impact that affects viral properties, without being located in structurally recognized domains (Table 2).

Among these mutations, 10 were found to be shared between different proteins, namely: PB1 and PB2 share 6 mutations (L13P, 398Q, G70, V504, P69, S42), PA and PB2 share 2 mutations (615R, 558T), and NA and PB2 share 2 mutations (H274Y, 122K).

The systematic review also highlighted significant serotypic diversity, reporting a total of 17 serotypes of avian influenza A virus: H5N1, H9N2, H7N4, H17N10, H18N11, H7N9, H2N2, H1N1, H3N2, H5N2, H5N8, H6N5, H7N7, H6N1, H7N3, H10N8, and H10N4.

### 3.2. Protein Sequence Dataset

The analysis conducted on the NCBI Virus database identified a total of 9002 protein sequences associated with the avian influenza virus type A (Tax ID 11320). The distribution among the ten selected host species is as follows: 1975 sequences from Homo sapiens (man), 1271 from Gallus gallus (chicken), 172 from Canis lupus familiaris (dog), 20 from Felis catus (domestic cat), 1511 from Sus scrofa (pig), 372 from Meleagris gallopavo (turkey), 22 from Cygnus (swan), 76 from Anser (goose), 55 from Laridae (gull) and 1435 from Anas (duck). The figure below illustrates the species distribution of the sequence mutation dataset used in our study (Figure 1).

From the totality of the protein sequences for each species, 464 sequences were selected and downloaded, representative of 58 viral isolates (Figure 2) belonging to 27 different serotypes (H5N2, H7N9, H5N8, H9N2, H10N8, H4N6, H5N1, H7N3, H3N2, H6N1, H5N6, H1N1, H1N2, H2N2, H7N8, H4N8, H12N8, H7N7, H13N2, H13N8, H13N6, H5N9, H13N9, H6N2, H3N8, H7N2, H7N1) (Table 3).

Amongst these, to better represent human sequences within the dataset and highlight their importance, 48 reference sequences (RefSeq) belonging to Homo sapiens were included. The data downloaded for the construction of the dataset, therefore, included protein sequences isolated from: RefSeq man (48), chicken (88), dog (16), domestic cat (16), pig (48), turkey (48), swan (8), goose (40), gull (40), and duck (112). The oldest sequence in the selected dataset is from 2009 and is related to a swan isolate, the only one available for this species. For the other species, the oldest date is 2013, confirming the choice of complete and recent sequences. Documented sources of isolation include the oro-nasopharynx, lungs, feces, and swabs. However, for 23 isolates the origin was not reported. Furthermore, the geographical distribution of the selected sequences includes a wide range of geographical areas, including the United States, China, the United Kingdom, Mexico, Taiwan, South Korea, Puerto Rico, Brazil, Hungary, and Chile (Figure 3).

### 3.3. Alignment and Phylogenetic Relationship Analysis

Multiple alignments of the reference human protein sequences (RefSeq) for the eight analyzed proteins yielded significant results for each of them. The PB2, PB1, PA, NP, M2, and NS1 proteins showed high identity percentages, generally above 90%, with almost complete query coverage.

Among these, the NS1 protein recorded slightly lower identity values than the others, with a maximum of 89.86%. For the PB2 protein, the query coverage reached 100% in all the analyzed sequences. The identity percentages ranged between 93.68% and 96.44%, while the E-values were equal to 0.0. Maximum scores ranged from 1500 to 1532, with the highest value associated with the A/Shanghai/1/2013(H7N9) strain and the lowest with the A/New York/392/2004(H3N2) strain.

The PB1 protein showed similarly high values, with a query coverage of 99%, an E-value of 0.0, and identity percentages ranging from 94.97% (A/California/07/2009(H1N1)) to 96.96% (A/Korea/426/1968(H2N2)). Maximum scores ranged from 1533 to 1554, with the highest value attributed to the sequence of the A/Korea/426/1968(H2N2) strain (Figure 4).

Sequence alignments of the PA protein show maximum scores ranging from 1395 to 1439, with the highest value found in the A/Shanghai/02/2013(H7N9) strain. The query coverage is 100% in most sequences, with identity percentages ranging from 92.18% to 95.25%. The maximum value was observed for the A/Shanghai/02/2013(H7N9) strain, while the minimum for the A/New York/392/2004(H3N2) strain.

For the NP protein, the maximum scores are between 959 and 1011, with a query coverage of 100% and an E-value of 0.0. The identity percentages range from 90.56% to 96.99%, with the highest value associated with the A/Shanghai/02/2013(H7N9) strain and the lowest with the A/New York/392/2004(H3N2) strain.

The M2 protein showed very low E-values, ranging from 1 × 10^−67^ (A/Shanghai/02/2013(H7N9)) to 8 × 10^−63^ (A/Korea/426/1968(H2N2)). The maximum scores ranged from 172 to 186, with a query coverage of 100% and identity percentages ranging from 83.51% to 90.72%, with the maximum value associated with the A/Shanghai/02/2013(H7N9) strain. The NS1 protein exhibits a higher variability, with identity percentages ranging from 78.70% to 89.86% in the A/California/07/2009(H1N1) and A/Korea/426/1968(H2N2) strains, respectively. Maximum scores range from 375 to 412, with 100% query coverage for most strains. E-values range from 1 × 10^−141^ to 3 × 10^−138^.

The second group of proteins, characterized by lower identity percentages, includes the HA and NA proteins. For the HA protein, the maximum scores range from a value of 430, recorded for the A/New York/392/2004(H3N2) strain, up to 604 for the A/Puerto Rico/8/1934(H1N1) strain. The query coverage ranges between 96% and 100% and most of the E-values were equal to 0.0. The identity percentages showed a certain variability, with a minimum of 40.21% for A/New York/392/2004(H3N2), and a maximum of 53.01% for A/Puerto Rico/8/1934(H1N1) (Figure 5).

Alignments for the NA protein yielded maximum scores ranging from 359 to 863. Again, query coverage was high, with values ranging from 99% to 100%. E-values were 0.0 for the A/Korea/426/1968(H2N2) and A/New York/392/2004(H3N2) strains, while for A/Shanghai/02/2013(H7N9), A/California/07/2009(H1N1), and A/Puerto Rico/8/1934(H1N1), they were 3 × 10^−131^, 4 × 10^−128^, and 4 × 10^−124^, respectively. The identity percentages show values ranging from a minimum of 42.98% for A/Puerto Rico/8/1934(H1N1) to a maximum of 87.42% for A/Korea/426/1968(H2N2).

The alignment of the protein sequences of the eight analyzed proteins, coming from the six human reference sequences, with the protein sequences of each strain belonging to the nine selected animal hosts, produced a total of 432 phylogenetic trees.

### 3.4. Evolutionary Proximity and Shared Mutations

After a detailed analysis of the evolutionary proximity of each sequence related to each protein, examining the 432 phylogenetic trees produced by the previous alignments, a total of 55 trees were selected (Figure 6). This selection represents the result of an evaluation aimed at identifying only the phylogenetic trees that show a significant evolutionary proximity between the reference human sequences and different hosts belonging to different serotypes. Of the total 55 trees selected for the subsequent evaluation on the presence of mutations, 92 pairs of sequences showing evolutionary proximity were identified. The 92 pairs of amino acid sequences selected belong to five of the eight proteins chosen during the review of the scientific literature. In detail, 15 pairs are related to the PB1 protein, 13 to PB2, 18 to PA, 11 to M2, and 35 to the NS1 protein (Figure 6). The other three proteins analyzed in the previous phases (HA, NA, and NP) were excluded from the final analysis for various reasons. In the case of the NP protein, the associated amino acid sequences did not meet the established evolutionary proximity criteria. For the HA and NA proteins, in addition to not demonstrating significant evolutionary proximity between different serotypes, any cases of proximity did not meet the requirement of a percentage of identity that was equal to or greater than 80%.

Of the 92 pairs of amino acid sequences compared, 75 showed the presence of mutations associated with spillover. Of these, 6 came from chicken, 6 from dog, 8 from cat, 14 from pig, 18 from turkey, 2 from goose, 7 from seagull, and 19 from duck.

In total, the analysis identified 30 amino acid mutations in the comparison between the host sequences and the human reference genomes, divided among the proteins as follows (Figure 7): 4 for PB1 (R691K, L13P, S375N, K52R), 4 for PB2 (K683T, V504, R389K, 714S), 20 for PA (E684G, P224S, L259P, I550L, P190S, V127, L672, G18E, R388S, E448A, S37A, N383D, L336M, Q400P, 615R, R204K, T85I, G186S, K356R, N409S), 1 for M2 (S31N), and 1 for NS1 (P42S).

All selected hosts showed mutations in their amino acid sequences, except for the swan, which did not show significant alignments. Mutations belonging to the host sequences showed the following distribution: 17 in chicken, 16 in dog, 19 in cat, 25 in pig, 17 in turkey, 4 in goose, 4 in seagull, and 20 in duck (Figure 8).

The viral strains that showed the greatest number of mutations came from chickens, dogs, cats, pigs, turkeys, seagulls, and common teals (Figure 9).

The A/chicken/TC/a504/2015(H5N2) virus presents mutations in the PB1 (R691K, L13P) and PB2 (K683T, V504, 714S) proteins (Figure 10).

Similarly, the A/chicken/Jalisco/CPA-01858-16-CENASA-95294/2016(H7N3) virus shows significant mutations in the PA protein (E684G, P224S, L259P, I550L, P190S, V127, L672, G18E, R388S, E448A, S37A) and the NS1 protein (P42S). Another poultry isolate, A/chicken/England/474-016250/2014(H4N6), has a similar genetic profile, with the same mutations found in PA and NS1. The A/canine/Taiwan/E01/2014(H6N1) virus has mutations in PB1 (R691K, L13P) and PA (E684G, L259P, I550L, N383D, P190S, V127, L672, E448A, S37A, L336M, G18E), while in the case of A/feline/Korea/FY028/2010(H3N2) mutations have been identified in the PA protein (E684G, P224S, L259P, I550L, N383D, P190S, Q400P, V127, 615R, L672, G18E, E448A, S37A) and in PB2 (K683T, V504, 714S). Another isolate, A/cat/Sichuan/SC18/2014(H5N6), shows mutations in PB1 (R691K, S375N, L13P) and PB2 (K683T, V504, 714S). A/swine/Shandong/SD1/2014(H5N1) virus shows more complex mutations, involving PB1 (R691K, L13P, S375N), PB2 (K683T, V504, 714S), PA (P224S, L259P, I550L, N383D, P190S, V127, L672, G18E, E448A, S37A), and NS1 (P42S). Similar mutations were also found in A/swine/Mexico/GtoDMZC02/2014(H5N2), with PB1 (R691K, L13P) and PA (P224S, L259P, I550L, N383D, P190S, V127, L672, G18E, E448A, S37A).

In the case of A/swine/Yantai/16/2012(H9N2), mutations were found in PB1 (L13P, K52R) and NS1 (P42S). A/turkey/Missouri/16-014037-7/2016(H5N1) virus has mutations in PA (E684G, P224S, L259P, I550L, N383D, P190S, Q400P, V127, L672, G18E, E448A, S37A) and NS1 (P42S). Another isolate, A/turkey/California/8199/2015(H7N3), shows mutations in PB2 (K683T, R389K, V504, 714S) and NS1 (P42S). The A/wild bird/Chile/1805/2008(H5N9) strain has mutations in PB2 (K683T, V504, 714S) and NS1 (P42S). The A/common teal/Nanji/NJ-101/2014(H9N2) virus has mutations in PB2 (K683T, R389K, V504, 714S), M2 (S31N) and NS1 (P42S), while A/common teal/Nanji/NJ-262/2013(H6N2) has mutations in PB1 (R691K) and PA (P224S, E684G, L259P, I550L, N383D, P190S, Q400P, V127, L672, G18E, E448A, S37A). Among all animal species, the strain “A/swine/Shandong/SD1/2014(H5N1)” showed the highest number of mutations (17 in total), related to the highest number of proteins (four out of five), in common with the human reference sequences (Figure 11).

## 4. Discussion

The present study highlights the fundamental role of amino acid mutations in the phenomenon of spillover of the avian influenza virus, analyzing 156 mutations distributed on eight viral proteins: PB2, PB1, PA, NP, HA, NA, M2, and NS1. The systematic review of the recent scientific literature, in addition to being preparatory for the bioinformatic analysis of the sequences, was an important tool to be able to identify within the data provided by the scientific community which and how many mutations are considered potentially responsible for the phenomenon of spillover. This knowledge could be used not only in the context of experimental analysis and evaluation but also applied in contexts of health risk assessment and management. Among these, the proteins PB1, PB2, and PA showed a high frequency of mutations shared between some animal species and the human reference sequences. The PB1 protein plays an essential role in viral replication, the mutations observed in different hosts act by optimizing the interaction with species-specific proteins [10]. The R691K, L13P, and S375N mutations were found in chickens, pigs, and cats, demonstrating their relevance in the context of interspecies transmission. The R691K mutation, being associated with an increase in replicative efficiency in mammalian cells [12], suggests that the substitution could optimize such replication in non-avian species. Located in the N-terminal domain, L13P introduces greater structural flexibility, improving interactions with PA and the stability of the polymerase complex [13]. This mutation, found in chickens, pigs, and ducks, is also associated with an increase in the replicative efficiency of the virus in mammalian cells, contributing to its interspecific adaptation. Another aspect of great interest is represented by the S375N mutation, observed in viral strains belonging to pigs and cats. This alteration has been found in strains that show an evolutionary proximity to human genomes and could be related to an improvement in the stability of the ribonucleoprotein complex (RNP) in different cellular environments, favoring the adaptation of the virus to new hosts [14]. Furthermore, the amino acid mutations detected in PB1 have shown a particular distribution among the analyzed serotypes. In the H5N1 and H7N9 strains, some mutations are associated with greater pathogenicity and ability to spread among mammals [15]. During the analysis, four mutations were identified in the PB2 protein (K683T, V504, R389K, 714S), most of them related to interspecific adaptation and replication efficiency. The K683T mutation, being close to the region involved in the binding of the host messenger RNA cap, could improve the ability of the polymerase to efficiently transcribe the viral RNA [16]. The V504 mutation also highlights a possible coordination mechanism between the proteins of the polymerase complex [17]. This alteration has been detected in strains from hosts such as chickens, pigs, and ducks, species already known for their role as reservoirs of zoonotic viruses. The R389K mutation, instead, has been found in strains isolated from turkeys and pigs. In this case, it should be emphasized that this substitution is in a critical region of the protein sequence implicated in the interaction of PB2 with other RNP proteins (PA and PB1) that increase its stability in host cells [18]. Another mutation in PB2 implicated in RNP binding and nuclear translocation is 714S, which has been detected in strains from chickens and pigs. Functionally, PB2 also interacts with specific host receptors. Although this study did not include silico structural analyses, the literature suggests that mutations such as K683T and 714S could influence the binding of the protein to nuclear receptors, enhancing the translocation of the polymerase complex into cells [16,19]. In addition, 20 mutations associated with the PA protein have been identified (E684G, P224S, L259P, I550L, P190S, V127, L672, G18E, R388S, E448A, S37A, N383D, L336M, Q400P, 615R, R204K, T85I, G186S, K356R, N409S), many of which are linked to spillover. Molecular interactions with other proteins and with the host cell that performs PA are optimized by these mutations. Among all, the most significant mutations were E684G, known to be associated with an increase in the efficiency of cap cleavage [20], present in pigs, turkeys, and cats; P224S, which could influence the binding to PB1 [21]; and the mutation L259P identified in chicken and pig isolates, which according to some studies, could increase the capacity of the virus to process the host messenger RNA, thus accelerating the viral replication cycle [22]. Furthermore, it is important to highlight the presence of mutations such as I550L and R388S. The first, frequently detected in turkey and pig strains, improves viral efficiency by acting on the binding affinity for cellular receptors [23], the second plays a fundamental role since it could improve the cooperation between PA and PB1, increasing the stability of the polymerase complex [24]. Mutations shared between human and animal sequences identified in the M2 and NS1 proteins should not be underestimated. For each of these proteins, only one mutation shared between different hosts was found, S31N in M2 and P42S in NS1, respectively. These mutations enhance cell survival and the ability to evade host immune defenses. The S31N substitution in the M2 protein has been found in strains from species such as ducks and gulls. It is important to highlight that the substitution of serine with asparagine in the viral proton channel confers increased resistance to antiviral drugs, such as amantadine [25]. The spread of S31N in zoonotic strains indicates that this alteration represents a favorable adaptation that could enhance the survival of the virus in new ecological contexts, increasing the probability of spillover. The P42S mutation in the NS1 protein, observed in chicken, pig, and turkey strains, induces a structural change in the functional region of the protein involved in the modulation of the host immune response [25]. The substitution of proline with a serine could improve the interaction of NS1 with cellular factors responsible for the innate antiviral response, thus enhancing the ability of the virus to evade host defense mechanisms. This mutation seems to optimize virulence and viral replication, providing the virus with a selective advantage during spillover. Phylogenetic and bioinformatic analysis of isolates from animal hosts coupled to human reference sequences showed that some of these strains present a considerable number of mutations shared with humans. Of experimental importance, and strengthening the hypotheses of this study, is the fact that these shared mutations in key proteins originate from pigs, chickens, ducks, and cats, animal species with significant zoonotic potential and, in some cases, with marked synanthropy. A prime example is the A/swine/Shandong/SD1/2014(H5N1) strain, isolated from pigs, which is characterized by the presence of 17 mutations distributed across four main proteins: PB1, PB2, PA, and NS1. In PB1, mutations such as L13P and R691K optimize the stability of the polymerase complex and increase the replicative efficiency of the virus. In PB2, alterations such as K683T and V504 favor the nuclear transport of the ribonucleoprotein complex, improving adaptation to new cellular substrates. The PA protein also displays significant mutations, including E684G, L259P, and P224S, which enhance nuclease activity, accelerating viral RNA synthesis. Finally, NS1 presents the P42S mutation, which enhances the evasion of the innate immune response. The combination of these mutations makes the A/swine/Shandong/SD1/2014(H5N1) strain a paradigmatic example of a highly adapted and potentially dangerous virus for humans, highlighting the role of the pig as a critical reservoir for genomic reassortment.

The A/chicken/TC/a504/2015(H5N2) strain, presents mutations in PB1 and PB2. This strain highlights how chickens can act as “evolutionary bridges”, facilitating genomic reassortment between avian strains and human-adapted variants.

The turkey emerges as another relevant host, as demonstrated by the A/turkey/Missouri/16-014037-7/2016(H5N1) strain. This isolate presents mutations in PB2, PA, and NS1 that, in this case, also enhance interactions with the polymerase complex and replication capacity. The role of turkey as an intermediate reservoir for zoonotic viruses is, therefore, highlighted by the presence of critical mutations. Among wild species, ducks remain a natural reservoir of avian influenza viruses, as demonstrated by the strain A/common teal/Nanji/NJ-101/2014(H9N2). This isolate presents significant mutations in PB2, M2, and NS1.

Species less commonly associated with these viruses, such as cats, have also shown strains with relevant mutations. The strain A/feline/Korea/FY028/2010(H3N2) presents mutations in PB1 and PB2, including L13P and R691K in PB1, which enhance viral replication, and K683T in PB2, which optimizes nuclear transport.

To achieve these results, the study was based on a predominantly bioinformatics approach. The NCBI Virus database and tools such as BlastP, coupled with phylogenetic construction algorithms, such as Fast Minimum Evolution, have allowed us to detect evolutionary relationships between animal and human sequences, producing 432 phylogenetic trees in total. The study aimed to highlight how the integration between evolutionary assessments and detection of key mutations is necessary to be able to highlight how these are distributed among hosts and provide molecular bases for understanding zoonotic risk. The identification of shared mutations between human and animal strains could be used to develop predictive mathematical models that quantify the risk of spillover, also supporting advanced genomic surveillance, potentially integrated into public health programs to monitor and prevent pandemic events. The use of this approach implies that the results must be contextualized, and all limitations must be evaluated.

One of the main limitations concerns the nature of the bioinformatics dataset, which reflects only the isolations available in the NCBI Virus database. These data, although representative, are often incomplete and lack crucial information, such as the ecological context or specific interactions between hosts. The lack of in vitro and in vivo experiments useful for validating the hypotheses formulated prevents us from confirming with certainty the functional role of mutations [26], which, although previously confirmed through studies published in the literature and reviewed in the first phase of the study, always remain limited to specific geographical contexts, without considering the interactions between species in transmission dynamics. The method used, although it evaluates the species most implicated in the transmission and epidemiology of the avian influenza virus, does not integrate tools capable of providing advanced epidemiological models. Furthermore, the experimental analysis of this study was limited to identifying the mutations known to the scientific community up to now. The limitations described, in addition to contextualizing the results, show how the prospects related to this study are broad and multidisciplinary. The next steps to this study must be linked to the in vitro and in vivo validation of mutations to better understand the molecular interactions between host and virus; this may also be necessary with the help of integrated omic sciences. Computational approaches, such as molecular docking, capable of predicting the structural changes caused by mutations and their implications in the binding affinity with the receptors of different hosts, help to better understand the molecular mechanisms underlying a potential spillover [27,28]. These tools, integrated with software capable of simulating structural variations due to changes in sequence and applying predictive algorithms for protein structure, allow us to obtain detailed models that illustrate how specific mutations influence the conformation of viral proteins. The improvement of the method of collecting and analyzing epidemiological data, thanks to the integration of sources from databases of national and international institutions, could be strategic in strengthening the ability to intervene preventively in emergency situations [29]. Another necessary development concerns the integration of the methods used in this study with mathematical models and epidemiological simulations, to be able to predict and quantify the risk of spillover, avoiding or managing in a rational way possible scenarios of diffusion of the virus [30]. In this context, it becomes essential to conduct more in-depth analyses of the frequency and distribution of mutations in different host species. Appropriate statistical tests can be useful in identifying significant trends in the presence of mutations and assessing interspecific variation [31].

Detecting repeated signals of adaptation linked to spillover through statistical analysis would improve the ability to interpret molecular surveillance data [32]. The mutations identified in this study can be used in the design and development of targeted vaccines that consider the amino acid alterations implicated in spillover. The study identified mutations in PB1, PB2, and PA that are conserved across species and may offer a unique opportunity for the design of universal vaccines. This, in pragmatic terms, means that by targeting highly conserved protein regions in hosts, the need for frequent updates could be reduced [33]. An application example could be the development of vaccines that exploit messenger RNA technology and include sequences coding for mutant proteins. The data collected from this study could, together with future studies, provide a knowledge base for the development of multivalent vaccines [34]. In contexts where proximity and interaction between animal species and humans are frequent, vaccines able to provide protection against a wide range of serotypes associated with spillover could represent a strategic prevention tool [35].

The knowledge base provided by this study may also open new scenarios in the development of advanced diagnostic kits. Detection of key mutations associated with spillover, using next-generation sequencing (NGS) technologies, could be strategic in managing health interventions or in understanding future transmission dynamics in advance [36,37]. Molecular diagnostic tools that detect the presence of mutations beyond the virus could find field applications in monitoring viral strains at high zoonotic risk [38]. In this sense, syndromic diagnostic kits could lead to a more complete risk assessment. Early detection of mutations associated with spillover would allow health authorities and others to intervene promptly through containment measures, preventing the spread of high-risk strains [39]. The central hypothesis of the study is that these mutations represent a molecular signature of the interspecific adaptation process, favoring the stability of the ribonucleoprotein complex, interaction with host proteins, and modulation of the host immune response. This view is also supported by the identification of conservation mutations in M2 and NS1, already known for their ability to evade immune defenses and confer resistance to antiviral drugs [40,41]. The evidence presented outlines a robust molecular framework for interpreting the mechanisms underlying influenza zoonosis and suggests the need to strengthen integrated genomic surveillance systems, develop diagnostic tools capable of detecting high-risk zoonotic mutations, and target vaccine design to highly conserved protein regions. Ultimately, the study highlights how the combined analysis of key mutations and phylogenetic trajectories of AIV viruses can provide an essential knowledge base for the prediction and prevention of future pandemics.

## 5. Conclusions

To achieve these results, the study was based on a predominantly bioinformatics approach. The NCBI Virus database and tools such as BlastP, coupled with phylogenetic construction algorithms, such as Fast Minimum Evolution, have allowed us to detect evolutionary relationships between animal and human sequences, producing 432 phylogenetic trees in total. The study aimed to highlight how the integration between evolutionary assessments and detection of key mutations is necessary, to be able to highlight how these are distributed among hosts and provide molecular bases for understanding zoonotic risk. The identification of shared mutations between human and animal strains could be used to develop predictive mathematical models that quantify the risk of spillover, also supporting advanced genomic surveillance, potentially integrated into public health programs to monitor and prevent pandemic events. The use of this approach implies contextualizing the results and evaluating all limitations, despite which, this study can provide a solid basis for further research and targeted interventions. In conclusion, the study conducted aims to represent an advance in the understanding of the molecular and evolutionary mechanisms associated with the avian influenza virus, providing a fundamental experimental basis for the development of more effective health interventions and for the design of advanced experimental studies.

## Figures and Tables

**Figure 1 pathogens-14-00608-f001:**
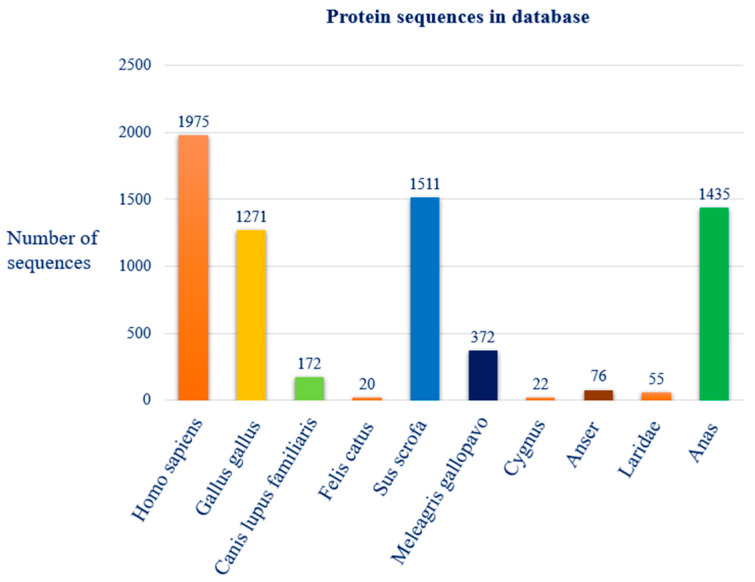
Distribution of protein sequences, among the selected species, present in the reference database “NCBI Virus”.

**Figure 2 pathogens-14-00608-f002:**
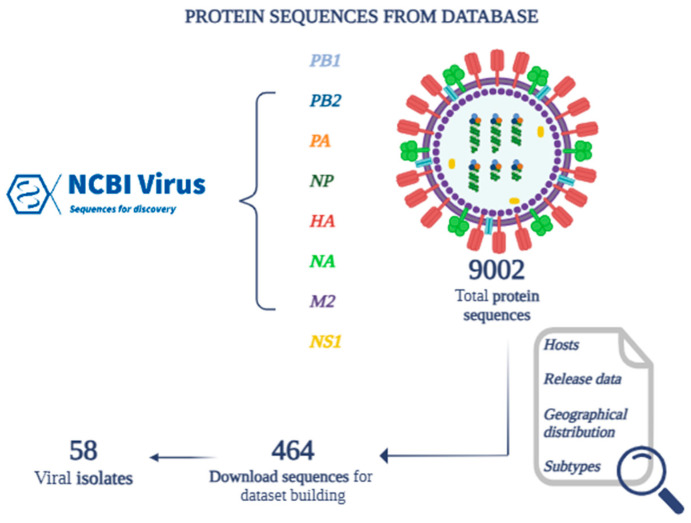
Building a dataset of protein sequences identified through the NCBI Virus database.

**Figure 3 pathogens-14-00608-f003:**
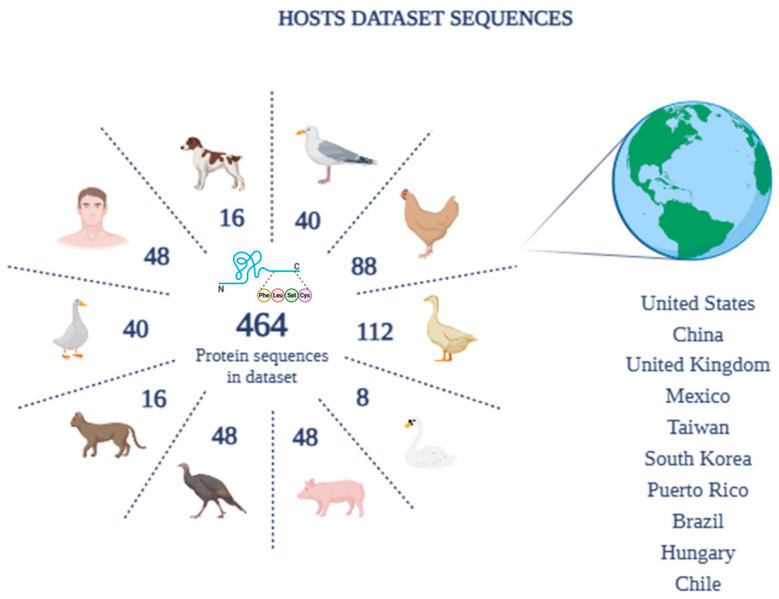
Complete sequence dataset by number of hosts and geographic distribution.

**Figure 4 pathogens-14-00608-f004:**
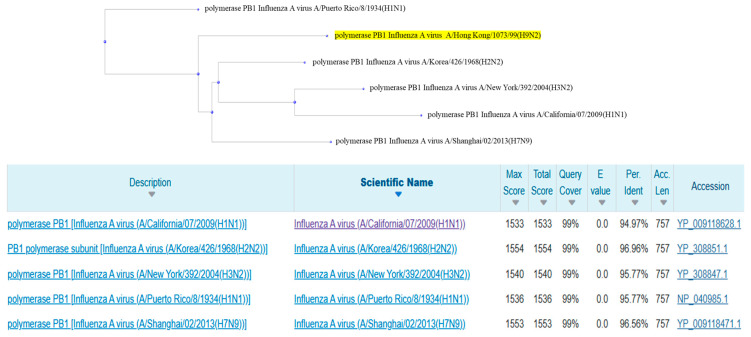
Phylogenetic tree and alignment results between the sequences related to the PB1 protein of all human reference sequences.

**Figure 5 pathogens-14-00608-f005:**
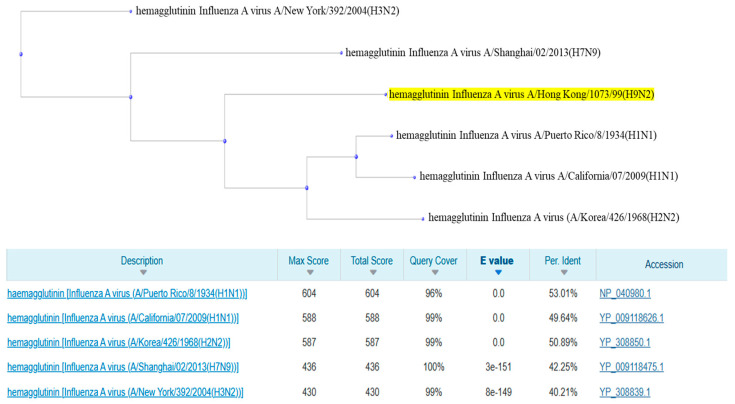
Phylogenetic tree and alignment results between the HA protein sequences of all human reference sequences.

**Figure 6 pathogens-14-00608-f006:**
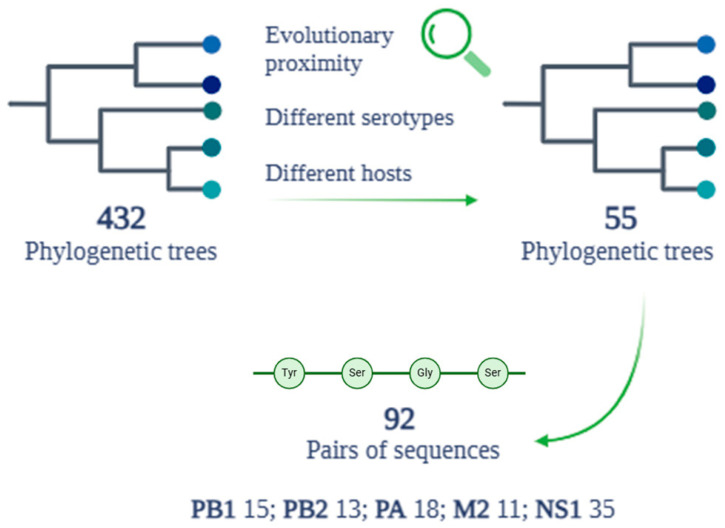
Phylogenetic tree and alignment results between the HA protein sequences of all human reference sequences.

**Figure 7 pathogens-14-00608-f007:**
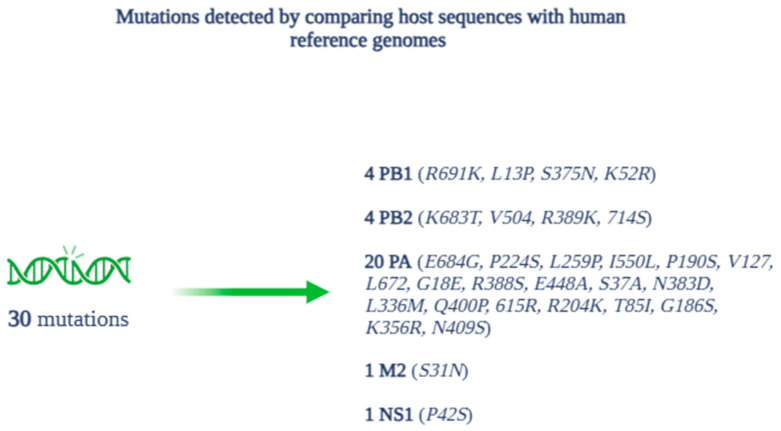
Mutations and related proteins detected during alignment and phylogenetic analysis.

**Figure 8 pathogens-14-00608-f008:**
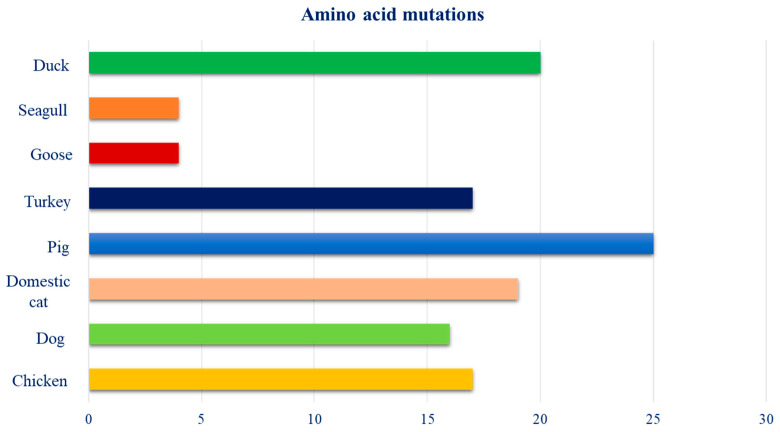
Distribution of amino acid mutations, among the selected species, present in the dataset created.

**Figure 9 pathogens-14-00608-f009:**
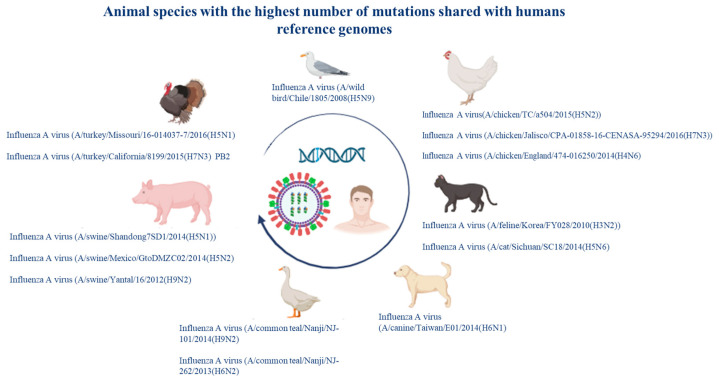
Viral strains in animal species that share the greatest number of mutations with the human reference sequences.

**Figure 10 pathogens-14-00608-f010:**
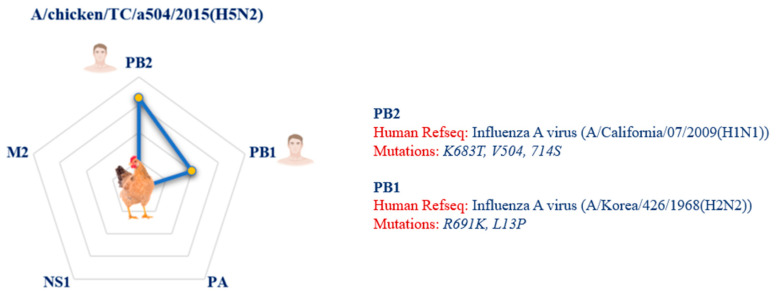
The image illustrates the relative proximity of proteins with amino acid mutations implicated in spillover, highlighting the viral strain isolated from chicken that shares the greatest number of mutations with the human reference sequences. The closeness in the radial plot to humans highlights the significant number of shared mutations. The legend next to the radial diagram shows, for each protein, the amino acid mutations shared between viral strains of animal origin and human strains. The human reference sequences (Human RefSeq) downloaded from the NCBI Virus database, with which the animal strains share these mutations, are also listed.

**Figure 11 pathogens-14-00608-f011:**
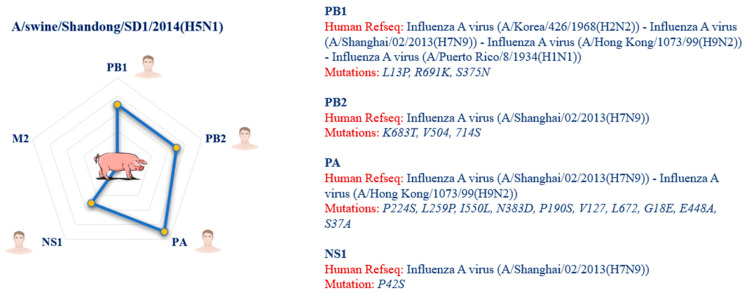
The image illustrates the relative proximity of proteins with amino acid mutations implicated in spillover, highlighting the viral strain isolated from swine that shares the greatest number of mutations with the human reference sequences. The closeness in the radial plot to humans highlights the significant number of shared mutations. The legend next to the radial diagram shows, for each protein, the amino acid mutations shared between viral strains of animal origin and human strains. The human reference sequences (Human RefSeq) downloaded from the NCBI Virus database, with which the animal strains share these mutations, are also listed.

**Table 1 pathogens-14-00608-t001:** Mutations reported in the scientific literature, associated with the related proteins and subtypes, potentially implicated in the spillover of influenza A virus.

Protein	Mutation	Serotypes
PB1	*H99Y*, *K577E*, *N47S*, *Q694H*, *I695K*, *R486K*, *V709I*, *E391*, *G581*, *T661*, *G580*, *A660*, *T296R*, *E180D*, *M317V*, *R691K*, *473R*, *L13P*, *398Q*, *G70*, *V504*, *P69*, *S42*, *S375N*, *K52R*, *L212V*	H5N1, H9N2, H7N4, H17N10, H18N11, H7N9, H2N2, H1N1, H3N2, H5N2, H5N8, H6N5
PB2	*E627K*, *G590S*, *Q591R*, *H274Y*, *I222K*, *D701N*, *K683T*, *S155N*, *K526R*, *I382S*, *R389K*, *E120D*, *V227I*, *S265*, *F406Y*, *Q591K*, *L386V*, *V649I*, *I66M*, *I109V*, *I133V*, *E158G*, *G591R/K*, *V504*, *615R*, *558T*, *714S*, *L13P*, *398Q*, *G70*, *P69*, *S42*, *A274T*, *E129K*, *K702R*, *E543D*, *A655Y*	H5N1, H9N2, H7N9, H7N4, H9N2, H17N10, H18N11, H7N7, H2N2, H1N1, H3N2, H5N2, H5N8, H6N5
PB2	*E627K*, *G590S*, *Q591R*, *H274Y*, *I222K*, *D701N*, *K683T*, *S155N*, *K526R*, *I382S*, *R389K*, *E120D*, *V227I*, *S265*, *F406Y*, *Q591K*, *L386V*, *V649I*, *I66M*, *I109V*, *I133V*, *E158G*, *G591R/K*, *V504*, *615R*, *558T*, *714S*, *L13P*, *398Q*, *G70*, *P69*, *S42*, *A274T*, *E129K*, *K702R*, *E543D*, *A655Y*	H5N1, H9N2, H7N9, H7N4, H9N2, H17N10, H18N11, H7N7, H2N2, H1N1, H3N2, H5N2, H5N8, H6N5
PA	*S388R*, *A448E*, *S49Y*, *D374G*, *T97I*, *I178M*, *M374T*, *V450A*, *E684G*, *F35L*, *K142E*, *P224S*, *L259P*, *I550L*, *T552S*, *M21I*, *S616P*, *E141K*, *N383D*, *M311I*, *R204K*, *P190S*, *Q400P*, *V127*, *L550*, *L627*, *615R*, *558T*, *L672*, *G18E*, *R388S*, *E448A*, *A36T*, *S37A*, *T85I*, *G186S*, *L336M*, *A343S/T*, *K356R*, *N409S*	H5N1, H7N9, H6N1, H9N2, H1N1, H5N2, H17N10, H18N11, H7N7, H7N3, H5N8
NP	*N319K*, *I109T*, *Q357K*, *105V*, *N52*	H5N1, H5N2, H1N1, H7N9
HA	*K165E*, *Q226L*, *N224K*, *G228S*, *V186G\K*, *K193T*, *A118T*, *S123N*, *A131V*, *R136K*, *L173I*, *M232I*, *H17*, *Y17*, *H106*, *H111*, *G225D*, *Q222L*, *G224S*, *G212R*, *N173H*, *N158D*, *T160A*, *T318I*, *H110Y*, *S138A*, *G186V*, *T221P*, *M66I*, *S141P*, *L322Q*	H1N1, H2N2, H3N2, H5N1, H6N1, H10N8, H7N9, H10N4, H9N2
NA	*L204M*, *G147R*, *Deleted region residue 69 to 73*, *H274Y*, *I222K*, *H275Y*, *I117T*, *E368K*, *S416G*, *A46D*, *S319F*, *S430G*	H3N2, H5N1, H7N9
M2	*V27A*, *V27T*, *S31G*, *S31N*	H5N1, H7N9
NS1	*P42S*, *I176T*, *K217T*	H5N1, H7N9, H5N2

**Table 2 pathogens-14-00608-t002:** Classification of mutations in avian influenza virus proteins based on their functional characteristics. Mutations are grouped into two categories: functional domain, which indicates substitutions located in well-characterized structural or functional regions of the protein; and functional impact, which refers to mutations that exert functional effects without being confined to a recognized structural domain. For each mutation, the affected viral protein, the type of classification, and a description of its potential role in viral replication or host adaptation are provided.

Mutation	Protein	Characteristic	Description
S388R	PA	Functional domain	Increases flexibility and associates with the vRNA promoter
A448E	PA	Functional domain	Forms hydrogen bonds with N444 interacting with PB1 and CDT POL II, enhancing the stability of PAC-C a16 helix
N409S	PA	Functional domain	Increases polymerase activity via interaction with PB-N
I382S	PB2	Functional domain	Amino acid substitution in the PB2 cap-binding domain
L386V	PB2	Functional domain	Mutation located within the cap-binding site
E190D, G225D	HA	Functional domain	Affects receptor-binding specificity (α2,3 SA, α2,6 SA)
G228S	HA	Functional domain	Forms a hydrogen bond with SA, increasing HA affinity for α2,6 SA
V186G-K193T-G228S/V186N-N224K-G228S	HA	Functional domain	Simultaneous substitutions for full α2,6 SA receptor specificity switch
N308S, A346V, T442A	NA	Functional impact	Residues near the active site of NA that may influence enzymatic activity
E368K, S416G	NA	Functional impact	Near the second sialic acid binding site, associated with neuraminidase activity and viral growth
K577E	PB1	Functional impact	Conformational change in PB1 α-helix potentially affecting binding affinity for PB2 α-helix
Q694H, I695K	PB1	Functional impact	Substitution in the PB1 C-terminal region
E627K	PB2	Functional impact	Recruits a second polymerase for nascent vRNP formation
Q226L	HA	Functional impact	Establishes a hydrophobic environment complementary to α2,6 SA C6 atom, favoring human receptor binding
G212R, N173H	HA	Functional impact	May involve the globular head epitope-binding site of HA
N158D, T160A	HA	Functional impact	Causes the loss of a glycosylation site near the receptor-binding site, increasing the preference for human receptors

**Table 3 pathogens-14-00608-t003:** Viral serotypes of the sequences present in the database associated with the different hosts.

Host	Serotypes
Human	H9N9, H3N2, H1N1, H2N2, H7N9
Chicken	H5N2, H7N9, H5N8, H9N2, H10N8, H4N6, H5N1, H7N3
Dog	H3N2, H6N1
Domestic cat	H5N6, H3N2
Swine	H1N1, H2N2, H3N2, H5N1, H5N2, H9N2
Turkey	H1N1, H2N2, H5N8, H5N2, H7N8, H7N3
Swan	H5N1
Goose	H4N8, H5N2, H5N6, H12N8, H7N7
Seagull	H13N2, H13N8, H13N6, H13N9, H5N9
Duck	H5N1, H5N2, H6N1, H6N2, H5N8, H3N8, H4N6, H7N1, H7N2, H7N3, H7N7, H13N6, H9N2

## Data Availability

The original contributions presented in this study are included in the article/Appendix A. Further inquiries can be directed to the corresponding author.

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
