# Peer review of "Avian Influenza Virus: Comparative Evolution as the Key for Predicting Host Tropism Expansion"

_pathogens, 2025, doi:10.3390/pathogens14070608_

Round 1
Reviewer 1 Report
Comments and Suggestions for Authors
The manuscript presents a comprehensive bioinformatics and phylogenetic analysis of avian influenza virus (AIV) evolution, focusing on amino acid mutations associated with host tropism expansion and spillover events. The study is timely and addresses a critical gap in understanding the molecular mechanisms underlying AIV adaptation to mammalian hosts, including humans. The methodology is robust, leveraging large-scale sequence data and phylogenetic tools, and the findings could inform surveillance and preventive strategies. However, several issues require clarification or improvement to enhance the manuscript’s impact and clarity.
1 The manuscript identifies 156 mutations across 8 proteins but does not explicitly justify why these specific mutations/proteins were prioritized. Were these mutations empirically validated in prior studies, or were they selected based on computational predictions? A table summarizing experimental evidence (e.g., in vitro/vivo studies) for each mutation’s functional impact would strengthen the rationale.
2 The phylogenetic analysis is descriptive but lacks statistical support (e.g., bootstrap values for tree nodes). Including confidence metrics for evolutionary proximity claims (e.g., "significant evolutionary proximity") is essential.
3 Were mutation frequencies compared across hosts using statistical tests (e.g., Fisher’s exact test)? Quantifying the significance of shared mutations would reinforce the conclusions.
4 While the study hypothesizes roles for mutations (e.g., PB2 K683T in replication efficiency), structural or mechanistic insights (e.g., molecular docking to predict binding affinity changes) are absent. Even a brief in silico analysis (e.g., using AlphaFold) could bridge this gap.
5Figure 1: The y-axis label is missing in the bar graph (presumably "Number of sequences").
6Table 1: Consider grouping mutations by functional domain (e.g., polymerase active site) to highlight patterns.
7Figures 10–11: Radial plots are innovative but unclear. Add legends explaining proximity metrics and human reference points.
Author Response
The Authors acknowledge the Reviewer for the revision of our manuscript. A point-to-point response is provided in the attachments

Reviewer 2 Report
Comments and Suggestions for Authors
In their article, “Avian influenza virus: comparative evolution as the key for predicting host tropism expansion,” the authors present a study that aimed to investigate the molecular mechanisms and evolutionary dynamics underlying the ability of influenza viruses to cross the species barrier and infect a broad spectrum of hosts, using a bioinformatics approach on viral sequences.
References should be described as follows, depending on the type of work:
1. Author 1, A.B.; Author 2, C.D. Title of the article. Abbreviated Journal Name Year, Volume, page range.
Introduction. In my opinion, the last paragraph is more proper for the discussion. Usually at the end of the Introduction authors present their aim.
Materials and Methods
“The hosts selected for the search were: human, chicken, dog, cat, pig, turkey, swan, goose, gull and duck.” I am wondering why you choose exactly these hosts? Are the ducks and geese domestic or wild? If they are domestic, why you include only gulls and swans; аre they synanthropic birds for your country?
Figure 3. Complete sequence dataset by number of hosts and geographic distribution – Why there are not included sequences from Europe?
When you made comments for the for the wild birds, it will be better to mention the exactly species. For example, lines 346-347 - “The viral strains that showed the greatest number of mutations came from chickens, dogs, cats, pigs, turkeys, seagulls and ducks (Figure 9)” For the “duck” in the figure 9 is mention Common teal.
Lines 501-559 – should be rewritten. No citations. Lines 501-512 – information is better suited for a conclusion. The purpose should be in the introduction. The limitations are well described, but there is text that is for the conclusion (549-599).
Author Response

(The authors gave the same response as above.)
